

# Comparison of 3D laser-based photonic scans and manual anthropometric measurements of body size and shape in a validation study of 123 young Swiss men

Nikola Koepke[1], Marcel Zwahlen[2], Jonathan C. Wells[3], Nicole Bender[1], Maciej Henneberg[1,4], Frank J. Rühli[1,*] and Kaspar Staub[1,*]

[1] Institute of Evolutionary Medicine, University of Zurich, Zurich, Switzerland
[2] Institute of Social and Preventive Medicine, University of Bern, Bern, Switzerland
[3] Childhood Nutrition Research Centre, Great Ormond Street Institute of Child Health, University College London, London, United Kingdom
[4] Adelaide Medical School, University of Adelaide, Adelaide, South Australia, Australia
[*] These authors contributed equally to this work.

Corresponding authors
Nikola Koepke,
nikola.koepke@uzh.ch
Kaspar Staub,
kaspar.staub@iem.uzh.ch

## ABSTRACT

**Background**. Manual anthropometric measurements are time-consuming and challenging to perform within acceptable intra- and inter-individual error margins in large studies. Three-dimensional (3D) laser body scanners provide a fast and precise alternative: within a few seconds the system produces a 3D image of the body topography and calculates some 150 standardised body size measurements.

**Objective**. The aim was to enhance the small number of existing validation studies and compare scan and manual techniques based on five selected measurements. We assessed the agreement between two repeated measurements within the two methods, analysed the direct agreement between the two methods, and explored the differences between the techniques when used in regressions assessing the effect of health related determinants on body shape indices.

**Methods**. We performed two repeated body scans on 123 volunteering young men using a Vitus Smart XXL body scanner. We manually measured height, waist, hip, buttock, and chest circumferences twice for each participant according to the WHO guidelines. The participants also filled in a basic questionnaire.

**Results**. Mean differences between the two scan measurements were smaller than between the two manual measurements, and precision as well as intra-class correlation coefficients were higher. Both techniques were strongly correlated. When comparing means between both techniques we found significant differences: Height was systematically shorter by 2.1 cm, whereas waist, hip and bust circumference measurements were larger in the scans by 1.17–4.37 cm. In consequence, body shape indices also became larger and the prevalence of overweight was greater when calculated from the scans. Between 4.1% and 7.3% of the probands changed risk category from normal to overweight when classified based on the scans. However, when employing regression analyses the two measurement techniques resulted in very similar coefficients, confidence intervals, and $p$-values.

**Conclusion**. For performing a large number of measurements in a large group of probands in a short time, body scans generally showed good feasibility, reliability, and validity in comparison to manual measurements. The systematic differences between the methods may result from their technical nature (contact vs. non-contact).

## INTRODUCTION

Body size and shape are directly linked to human health: being both underweight or overweight/obese is associated with increased morbidity and mortality risks (*Engeland et al., 2003*; *World Health Organization (WHO), 2009*; *Faeh, Braun & Bopp, 2012*; *Flegal et al., 2013*; *Kit, Ogden & Flegal, 2014*).

The prevalence of overweight and obesity (OW/OB) is still on the rise worldwide (*NCD-RisC, 2016*). Switzerland is no exception; OW/OB numbers have increased since the 1990s with significant variation by sex, age, region, migration status, and/or socio-economic background (*Faeh, Braun & Bopp, 2012*). Monitoring OW/OB on the population and/or individual level and identifying groups at risk is essential to plan public health interventions in order to reduce future health and economic burdens at the societal level (*Schmid et al., 2005*; *Schneider et al., 2010*; *Davin et al., 2012*; *Schneider & Venetz, 2014*).

Adipose tissue distribution has proven to be particularly important regarding the health risks associated with OW/OB. It is particularly abdominal fat that has adverse outcomes (cardiovascular diseases, type 2 diabetes, cancer, metabolic syndrome) (*Yusuf et al., 2005*; *Pischon et al., 2008*; *Gracia-Marco et al., 2016*). The standard imaging methods used to precisely assess body composition and fat distribution (MRI, DEXA, CT) are either invasive or time-consuming, and expensive, and thus not suitable for large-scale monitoring studies (*Machann et al., 2005*; *Lee & Gallagher, 2008*). A cheaper and faster way is the external assessment of body size and shape by standard anthropometric measurements using hand-held tape measures, callipers, and stadiometer, which are traditionally used in larger epidemiological studies (*Preedy, 2012*; *Malatesta, 2013*).

There is ongoing debate about the appropriate anthropometric proxy for OW/OB: Body mass index (BMI) is the most commonly used indicator, but it is not an ideal measure of body composition and fat distribution since it does not precisely differentiate between weight associated with lean/muscle mass and weight associated with fat mass (*Keys et al., 1972*; *Finucane et al., 2011*; *Malatesta, 2013*; *World Health Organization (WHO), 2004*). Even when BMI is well correlated with the percentage of body fat on the population level, there are exceptions and misclassifications at the individual level (most notably athletes) (*Marques-Vidal et al., 2008*; *Schneider et al., 2010*). Body shape indices, such as waist circumference (WC), waist-to-hip-ratio (WHR), or waist-to-height-ratio (WHtR) are recently gaining importance because of their improved indication of abdominal fat

(*Ashwell & Hsieh, 2005*; *Yusuf et al., 2005*; *Ogna et al., 2014*). Even when standardising guidelines and operation procedures (on posture, breathing position, tape-positioning, and tape-tension) exist and the measurements are performed by trained and experienced personnel, obtaining reliable, precise, and accurate manual waist and hip circumference is time-consuming in larger samples and challenging to perform within acceptable intra- and inter-individual errors (*Higgins & Comuzzie, 2012*; *Verweij et al., 2013*).

Recently, 3D photonic surface scans rapidly emerged as an attractive digital alternative to anthropometrically assessed body size and shape (*Lin et al., 2004*; *Olivares et al., 2007*; *Treleaven & Wells, 2007*; *Veitch, Veitch & Henneberg, 2007*; *Wells et al., 2008*; *Bretschneider et al., 2009*; *Wells, 2012*; *Olds et al., 2013*; *Jaeschke, Steinbrecher & Pischon, 2015*; *Loeffler et al., 2015*; *Peyer, Morris & Sellers, 2015*; *Wells et al., 2015*; *Kuehnapfel et al., 2016*). Within only 12–15 s the system scans the body surface; software algorithms then produce a 3D image of the body topography (consisting of 700,000 to one million individual data points in the 3D space), a large number of landmarks are automatically located by the software, and ca. 150 standard body size measurements are calculated from these scan landmarks (*Wells, 2012*). The advantage of 3D technology lies in the large number of measurements taken in a very short time, the improved standardisation due to the reduced physical contact between observers and probands (by eliminating potential error sources such as tape-tension, etc.), the possibility to adjust or take additional measurements (or composite new models for body shape) from the virtual body, and the electronic archiving of the 3D images for re-assessment and individual follow-ups with improved software (*Wells et al., 2015*). There are even some studies showing that 3-D body scan results correlate with total and regional fat mass (*Wang et al., 2006*; *Lee et al., 2015a*; *Lee et al., 2015b*; *Ng et al., 2016*), but further evidence is needed as to whether they can produce reliable predictions when compared to DEXA or MRI standards.

Thus, the scan technique—originally developed for the clothing industry—allows the comprehensive measurement of body shape in a large number of individuals in a short time (*Wells et al., 2015*). Consequently, its application in larger epidemiological studies on adults and children is currently gaining importance. A small number of recent validation studies comparing the scan technique with manual measurements (e.g., waist or hip circumferences) have shown the applicability of the scan technique in an epidemiological setting (*Wells, Treleaven & Charoensiriwath, 2011*; *Jaeschke, Steinbrecher & Pischon, 2015*; *Loeffler et al., 2015*; *Kuehnapfel et al., 2016*), using the same device as the one studied here. The scans generally showed good feasibility, reliability, and validity; correlations to parameters of the metabolic syndrome were similar to those of manual measurements (*Jaeschke, Steinbrecher & Pischon, 2015*). Scan and manual data showed generally a high level of ranking consistency and correlation between methods, but there are also indications for systematic bias (the scans provide slightly larger values than the manual measurements) (*Wang et al., 2006*; *Heuberger, Domina & Macgillivray, 2008*; *Choi & Ashdown, 2011*; *Jaeschke, Steinbrecher & Pischon, 2015*). Because there are different types of scanner hardware available, and also different ways through which software processes raw data to give final outputs, it is worth contributing additional validation studies.

**Table 1  Descriptive statistics of the study group ($N = 123$, all men) assessed by a questionnaire prior to the measurements.** Weight was separately measured in parallel to the scan and manual measurements, $N = 1$ weight measurement was missing. Height and BMI are reported in Tables 2 and 3.

| | Mean (SD) | Min | Max | N | Categories | Freq. | % |
|---|---|---|---|---|---|---|---|
| Weight (kg) | 73.96 (9.47) | 55.5 | 98.3 | 122 | | | |
| Age (years) | 24.55 (4.18) | 18 | 38 | 123 | Not reported | 2 | 1.6 |
| | | | | | 15–19 | 5 | 4.1 |
| | | | | | 20–24 | 69 | 56.1 |
| | | | | | 25–29 | 32 | 26.0 |
| | | | | | 30–34 | 12 | 9.8 |
| | | | | | 35–40 | 3 | 2.4 |
| Weekly hours of sports | 4.2 (3.0) | 0 | 17 | 123 | 0.0–0.5 | 7 | 5.7 |
| | | | | | 0.6–2.4 | 35 | 28.5 |
| | | | | | 2.5–4.9 | 37 | 30.1 |
| | | | | | 5.0–7.4 | 30 | 24.4 |
| | | | | | $\geq 7.5$ | 14 | 11.4 |
| Numbers of visits to a physician during the last year | 1.78 (0.82) | 1 | 5 | 123 | 1 | 51 | 41.5 |
| | | | | | 2 | 53 | 43.1 |
| | | | | | 3 | 15 | 12.2 |
| | | | | | $\geq 4$ | 4 | 3.3 |

The aim of this study was to compare the scanning and the manual anthropometric measurement techniques based on five selected body measurements (height, waist, hip, buttock, and chest circumferences) to add to the still small number of reliability and validation studies with the aim to further implement this method in population-based studies.

## DATA AND METHODS

### Study sample

The cross-sectional study was conducted in February 2013 on Irchel Campus of the University of Zurich. We collected scanned and manually measured anthropometric as well as questionnaire data on simple health-related aspects of 127 young adult males, voluntarily recruited via public advertisement (flyers, internet, social media, radio, oral communication, etc.) in the university environment. We assume that most of the probands were university students but we did not ask for occupation in the questionnaire. We exclusively focussed on young males because one of the future application of the technique will be among male Swiss Armed Forces conscripts. We excluded $N = 1$ (0.8%) participants due to age older than 40 years and $N = 3$ (2.4%) participants due to incomplete data. The remaining study sample ($N = 123$) was between 18 and 38 years old (mean age $= 24.55$ years, SD $= 4.18$), with 15 men (12.2%) being $\geq 30$ years (Table 1). The study sample included both, young men not regularly performing sports (34.2% $< 2.5$ h per week) and sportive young men (35.8% regularly performed $\geq 5.0$ h of sports per week). The number of participants with three or more visits to a physician during the last year was rather low (15.5%),

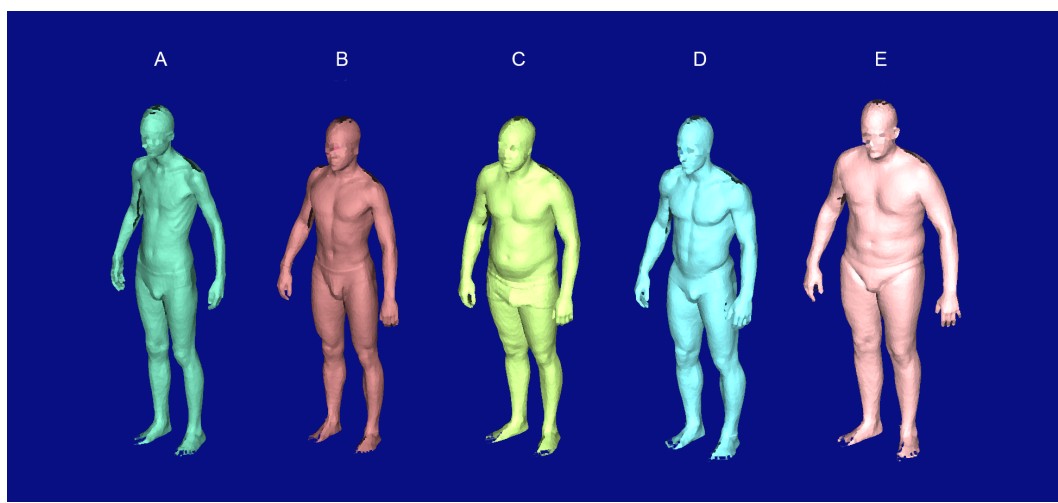

**Figure 1** **Raw scan outputs of five selected test subjects showing the full range of observed body shapes.** Subject (A) was the thinnest (scanned BMI = 16.85 kg/m$^2$) and subject (E) was the heaviest (BMI = 29.48 kg/m$^2$). Subject (B) represented the "healthy" body shape type with a BMI of 20.95 kg/m$^2$. Subjects (C and D) had a similar BMI (27.94 vs. 27.73 kg/m$^2$), but in contrast to subject (C), subject (D) represented the athletic body shape type (reporting 17 h of sport per week). The faces of the subjects have been pixelated and anonymised.

which indicates that most of the study probands were in good general health. Overall, the participants covered a wide spectrum of existing body shape variation in young men. The low-quality raw scan outputs (Fig. 1) show the thinnest ((A), scanned BMI = 16.85 kg/m$^2$) and the heaviest ((E), BMI = 29.48 kg/m$^2$) participant, as well as exemplary two participants, whose BMI was similar (27.94 vs. 27.73 kg/m$^2$), but in contrast to subject (C) subject (D) represented the more athletic body shape type (reporting 17 h of sport per week).

### Data collection

The participants underwent three measurement steps in separate rooms within a total maximum time of 20 min each: (a) filling in a basic questionnaire, (b) scan measurements (scans), and (c) manual measurements (hand/tape measurements).

### (a) Questionnaire data

In the first step, the participants filled in a basic questionnaire about their age (rounded to full years), their current physical activity level (expressed as total hours of sports per week), and the number of visits to a physician during the last year (as a crude proxy for their health status) (Table 1).

### (b) Scan measurements

In the second step scan measurements (SM = 3D body scans) were performed on the participants using a mobile Body Scanner *Vitus Smart XXL* and the software AnthroScan Professional (both from Human Solutions GmbH, Kaiserslautern, Germany), both integrated into a converted truck (the scanliner) of the Swiss Armed Forces Logistics Organisation (Armasuisse). The same Vitus Smart XXL scanner device is currently successfully used in different epidemiological studies in Germany (*Bretschneider et al.,*

*2009*; *Jaeschke, Steinbrecher & Pischon, 2015*; *Loeffler et al., 2015*; *Kuehnapfel et al., 2016*).
Each participant was subsequently scanned twice (scan measurements 1 and 2: SM1
and SM2). The participants wore form-fitting underpants only (without shoes) and a
tight-fitting bathing cap. The scans were performed by the study authors together with
experienced Armasuisse scan operators. The scanner was calibrated once at the beginning of
every measurement day following official protocols of the manufacturer. The participants
were asked to hold their breath in an exhaled status and stand in the standard position
defined by the manufacturer (standing upright, relaxed, on a marker setting, legs hip-wide
apart, upper limbs slightly bend at the elbows and held away from the body, head positioned
in accordance with the Frankfurt Horizontal Plane). Instructions for positioning on the
scanner platform were repeated prior to each scan. The scans were immediately inspected
visually for their quality, and had to be repeated in four out of a total of 254 scans (mostly
due to twisted topography in the head area caused by movement during the scan procedure).
Using four eye-safe lasers and eight cameras, the 3D-scanner provides a point cloud of about
one million data points, based on optical triangulation. In the standard setting, 154 digital
outcomes were determined by the software according to ISO 20685:2005 (*International
Organization for Standardization (ISO), 2005*). In order to focus on the shape of the trunk
(and thus abdominal fat) in future studies, we chose body height and four circumference
measurements of the trunk area to compare with the manual measurements. According
to the operating scan protocol of the manufacturer (automatic landmark positioning), we
chose the scan measurements "waist girth," "hip girth," "buttock girth," and "chest girth"
(across the bust point landmarks at the level of the nipples) to compare the scan technique
with the traditional manual measurements.

## (c) Manual measurements

During the third examination step each participant was manually measured twice by two
different trained anthropometrists. In $N = 1$ no measurement could be conducted. The
participants wore form-fitting underpants only (without shoes) and stood upright with
arms at the sides, feet positioned close together, and weight evenly distributed across the
feet. We measured height, waist circumference, hip circumference, buttock circumference,
and bust circumference twice for each participant (manual measurements 1 and 2: MM1
and MM2). Height was measured by a standard stadiometer (seca 216, fixed at the wall and
regularly calibrated/controlled during the measurement period). For the circumference
measurements a standard stretch-resistant hand-held tape was used and measurements
were taken at the end of normal expiration, with the tape parallel to the floor (according
to the official measurement protocols in use): Waist circumference was measured at the
approximate midpoint between the lower margin of the last palpable rib and the top of the
iliac crest (*World Health Organization (WHO), 2011*). Hip circumference was measured
around the widest portion of the buttocks (*World Health Organization (WHO), 2011*).
Buttock circumference was measured on the midpoint buttock curve (*NHANES, 1988*).
Chest circumference was measured at the level of the papilla of the breast. In addition,
the participants (wearing form-fitting underpants only) were weighed once to the nearest

0.1 kg with a standard medical weight scale (SECA 877 digital scale, Hamburg, Germany) during step (c).

## Body shape indices

Based on the mean of the two scans (mSM = (SM1 + SM2)/2) and the mean of the two manual (mMM = (MM1 + MM2)/2) measurements we calculated the most commonly used indices for body shape and abdominal fat distribution for each of the two techniques: BMI (= weight (kg)/(height (m$^2$)), WHR (= waist circumference/hip circumference), and WHtR (= waist circumference/height). In a next step, the participants were assigned to the official WHO risk categories for OW/OB for each of the indices and the two techniques (*Schneider et al., 2010*; *World Health Organization (WHO), 2004*).

## Statistical methods

First, we assessed the agreement between two repeated measurements within the two methods. To test the repeatability and the agreement between the repeated measurements within each of the two methods (SM1–SM2 and MM1–MM2) we calculated mean differences, intraclass correlation coefficients (ICC), precision (calculated based on the square root of the sum of squared differences between measurements divided by the number of observations) (*Altman, 1999*), and paired $t$-tests for two samples. For the direct comparison between scan and manual measurements we used the mean scan measurement (mSM) and the mean manual measurement (mMM).

Second, we analysed the direct agreement between scan and manual measurements for each of the selected measurements and for the resultant body shape indices to control for potential systematic differences. We calculated mean differences (mSM–mMM), correlation coefficients, and paired $t$-tests for two samples. Furthermore, we also performed Lin's concordance correlation coefficient (CCC) (*Lin, 1989*): it is calculated as CCC $= r *$ $C\_b$, where $r$ is the Pearson correlation coefficient and $C\_b$ is a bias correction factor. The $r$ variable evaluates precision, whereas $C\_b$ evaluates accuracy concerning the degree to which measures deviate from the best-fit line.

Visual inspection was done via scatterplots, kernel density plots, and the Bland-Altman (BA) method (*Bland & Altman, 1986*; *Bland & Altman, 1999*) for both intra- and inter-method comparisons. The BA method analyses the paired difference as a function of the average of the paired measurements. This is first done by graphical displays of the difference between the methods against their mean, then by calculating the mean difference and the standard deviation of the differences, which defines a range of agreement (*Jaeschke, Steinbrecher & Pischon, 2015*; *Wells et al., 2015*).

For the calculated body shape indices WC, WHR, and WHtR, we computed mean differences (mSM–mMM), Pearson correlation coefficients, Lin's concordance correlation coefficient (CCC), and paired $t$-tests for two samples to test the agreement between scan and manualf measurements. Reclassification between the official OW/OB categories when comparing the two methods was assessed by percentages in relation to the total sample and agreement by calculating kappa coefficients (*Altman, 1999*): Kappa was assessed according to Altman's reference range: 0.41–0.60 = moderate agreement; 0.61–0.80 = good agreement, >0.80 = excellent agreement.

**Table 2  Agreement within methods.** Reliability tests in-between the two scan measurements (scans, SM1 and SM2, $N = 123$) and in-between the two manual measurements (tape measurements, MM1 and MM2, $N = 122$).

| | | | Scan Measurement (SM) | | | |
|---|---|---|---|---|---|---|
| Measure (cm) | SM1 (SD) | SM2 (SD) | SM1–SM2 (SD) | ICC (95% CI) | Precision (cm) | Sig. paired $t$-test ($p$) |
| Height | 178.29 (6.54) | 178.33 (6.50) | −0.04 (0.55) | 0.998 (0.997–0.999) | 0.45 | 0.413 |
| Chest | 97.67 (6.48) | 97.56 (6.57) | 0.11 (1.78) | 0.981 (0.973–0.987) | 1.24 | 0.486 |
| Waist | 81.42 (6.86) | 81.34 (6.90) | 0.09 (1.16) | 0.993 (0.990–0.995) | 0.98 | 0.397 |
| Buttock | 97.28 (5.39) | 97.17 (5.45) | 0.11 (0.60) | 0.997 (0.995–0.998) | 1.18 | 0.052 |
| Hip | 99.19 (5.74) | 99.11 (5.87) | 0.09 (0.89) | 0.994 (0.992–0.996) | 1.05 | 0.280 |

| | | | Manual Measurement (MM) | | | |
|---|---|---|---|---|---|---|
| Measure (cm) | MM1 (SD) | MM2 (SD) | MM1–MM2 (SD) | ICC (95% CI) | Precision (cm) | Sig. paired $t$-test ($p$) |
| Height | 180.34 (6.56) | 180.29 (6.53) | 0.05 (0.42) | 0.999 (0.999–0.999) | 0.50 | 0.235 |
| Chest | 93.44 (6.19) | 94.19 (5.95) | −0.75 (2.01) | 0.968 (0.948–0.980) | 8.19 | <0.001 |
| Waist | 80.11 (6.12) | 80.50 (6.16) | −0.40 (1.19) | 0.990 (0.984–0.993) | 4.36 | <0.001 |
| Buttock | 84.30 (7.05) | 84.92 (6.99) | −0.62 (2.87) | 0.955 (0.935–0.969) | 6.84 | 0.018 |
| Hip | 94.67 (5.73) | 94.90 (5.68) | −0.23 (1.88) | 0.972 (0.960–0.980) | 2.50 | 0.186 |

Third, to assess if the two techniques result in similar results and coefficients when analysing sub-group differences, regression models testing health-related determinants of body shape indices based on questionnaire answers, we performed linear regressions with WC, WHR, and WHtR as dependent variables and categorised age, total hours of sports per week, and number of visits to a physician during the last year as independent variables. We used SPSS23 and Stata14 for all analyses.

## Ethics statement

All of the young men participated voluntarily in this study. They were informed in detail by the study leaders about the study (aim, protocol, risks, etc.) in written form prior to the measurement days and once again orally at the beginning of their measurements by the study leaders. Prior to the start of the measurements all participants signed a detailed informed consent form. At the time of measurement (February 2013) a waiver by the ethics committee of the Canton of Zurich was not needed for such non-clinical and non-invasive studies.

## RESULTS

### Agreement between the repeated measurements within methods

Agreement between the two repeated measurements within the two methods is reported in Table 2. The mean differences between the two scan measurements (SM1–SM2) were the smallest for height (−0.04 cm) and the largest for chest and buttock circumferences (0.11 cm). The ICC was smaller than 0.993 only in the case of chest circumference (0.981) and precision was below 1.24 cm for all SM. The paired $t$-tests for two samples resulted in non-significant $p$-values in all five measurements. The mean difference between the two manual measurements (MM1–MM2) was again the lowest (and non-significant) for height (0.05 cm) but significantly larger than −0.23 cm and up to −0.75 cm in all other manual measurements. The ICC between the two manual measurements was greater than

0.991 only in the case of height (0.999). Precision was higher than 2.50 cm and up to 8.19 cm in all manual measurements (indicating greater disagreement). Only in the case of height, the two manual measurements produced similarly low disagreeing results as the two repeated scan measurements (−0.04 cm vs. 0.05 cm mean difference, 0.998 vs. 0.999 ICC, 0.45 vs. 0.50 cm precision). The kernel density plots and BA plots confirm the generally closer agreement and higher reliability between the two scans than between the two manual measurements in the other four analysed measures (chest, waist, hip and buttock circumference) (Fig. S1).

## Agreement between methods: scan versus manual measurements

For the direct comparison between scan and manual measurements we calculated the mean of each of the two basic measurements, the mean scan measurement (mSM = (SM1 + SM2)/2) and the mean manual measurement (mMM = (MM1 + MM2)/2). Overall, correlation between mSM and mMM values was high ($r \geq 0.93$), except in the case of the buttock circumference ($r = 0.83$) (Table 3). When calculating Lin's concordance correlation coefficient (CCC), which adjusts Pearson's $r$ by a bias correction factor, CCC remains high ($>0.94$) for height and waist circumference. For chest circumference, CCC was 0.781 for hip circumference and 0.784 for chest circumference. For buttock circumference $C\_b$ was only 0.312, which resulted in a low CCC of 0.258 (Table 3). However, comparing mean levels revealed significant differences between mSM and mMM. In the case of height, mSM (178.31 cm, SD = 6.51) was significantly shorter than mMM (180.32 cm, SD = 6.54) by −2.01 cm (SD = 0.77, $p < 0.001$). In contrast, mSM was significantly larger than mMM in all analysed circumference measurements. The difference was the smallest for waist circumference (+1.17 cm, SD = 0.13) and the largest for buttock circumference (+12.62 cm, SD = 0.35). For chest and hip circumference the difference was in the expected range of approximately 4 cm, with +3.88 cm (SD = 2.17) for chest and +4.37 cm (SD = 0.19) for hip. The scatterplots, kernel density plots, and BA plots in Fig. 2 confirm the observed pattern: mSM were smaller than mMM for height and larger for the four circumference measures (chest, waist, hip, buttock). The kernel density plots further show that although the position of the distributions on the $x$-axis differed, their shape was consistently similar between mSM and mMM.

## Body shape indices: agreement between scan and manual measurements

We also tested the agreement between the two methods when using mSM and mMM to calculate the most commonly used indices of body shape—WC, WHR, WHtR, BMI—and when assigning the subjects to the official WHO categories regarding the linked health risk. The results for waist circumference (WC) are reported in the previous section. The correlation coefficients were very high (0.98. and 0.99) for WHtR and BMI, but lower (0.85) for WHR (Table 3). The concordance correlation coefficients (CCC) were high ($>0.92$) for WHtR and BMI, whereas the lower accuracy measure $C\_b = 0.786$ for WHR resulted in lower CCC of 0.673. The mean difference was −0.03 (SD = 0.002) for WHR. The scanned WHtR were on average 0.01 (SD = 0.001) higher than the manual measured WHtR,

**Table 3  Agreement and differences between the two methods.** Comparing the mean of the two scan measurements (mSM, $N = 123$) and mean of the two manual measurements (mMM, $N = 122$). CCC, Concordance Correlation Coefficient (CCC $= r * C\_b$).

| Measure (cm) | mSM (SD) | mMM (SD) | mSM–mMM (SD) | Sig. paired $t$-test ($p$) | CCC | Correlation ($r$) | $C\_b$ |
|---|---|---|---|---|---|---|---|
| Height | 178.31 (6.51) | 180.32 (6.54) | −2.01 (0.77) | <0.001 | 0.948 | 0.993 | 0.955 |
| Chest | 97.62 (6.47) | 93.82 (5.98) | 3.88 (2.17) | <0.001 | 0.784 | 0.941 | 0.833 |
| Waist (WC) | 81.38 (6.86) | 80.31 (6.11) | 1.17 (0.13) | <0.001 | 0.960 | 0.982 | 0.978 |
| Buttock | 97.23 (5.41) | 84.61 (6.87) | 12.62 (0.35) | <0.001 | 0.258 | 0.828 | 0.312 |
| Hip | 99.19 (5.79) | 94.77 (5.62) | 4.37 (0.19) | <0.001 | 0.718 | 0.933 | 0.769 |

| Index | mSM (SD) | mMM (SD) | mSM-mMM (SD) | Sig. paired $t$-test ($p$) | CCC | Correlation ($r$) | $C\_b$ |
|---|---|---|---|---|---|---|---|
| WHR | 0.82 (0.04) | 0.85 (0.03) | −0.03 (0.002) | <0.001 | 0.673 | 0.857 | 0.786 |
| WHtR | 0.46 (0.04) | 0.45 (0.03) | 0.01 (0.001) | <0.001 | 0.920 | 0.979 | 0.939 |
| BMI (kg/m$^2$) | 23.23 (2.44) | 22.72 (2.38) | 0.52 (0.210) | <0.001 | 0.974 | 0.996 | 0.977 |

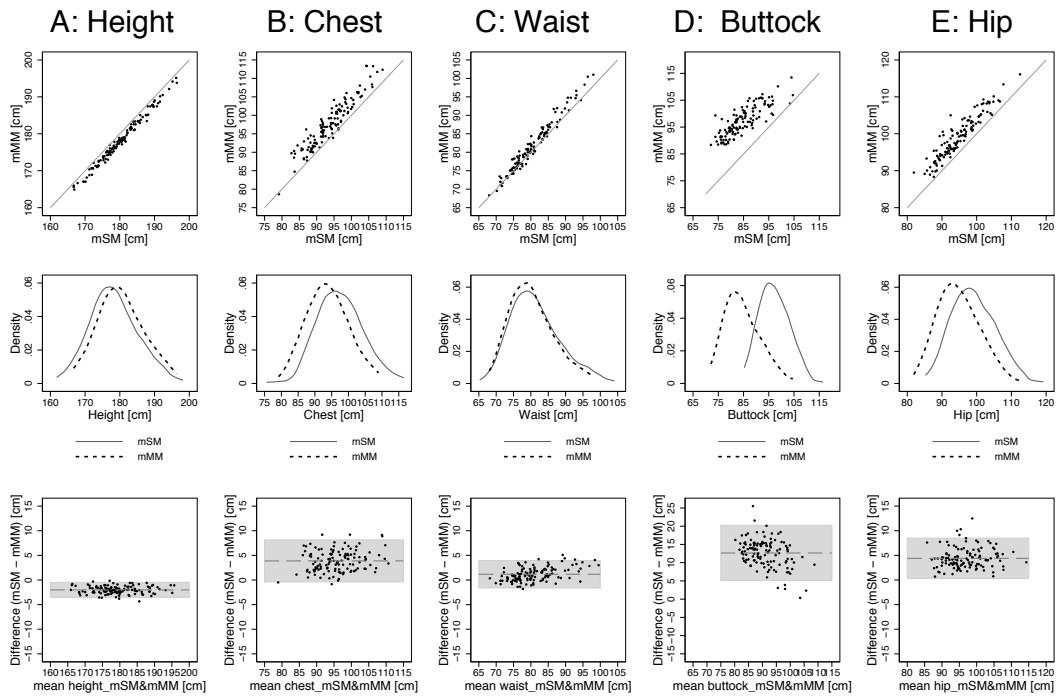

**Figure 2  Agreement between methods: mSM vs. mMM by scatterplots, kernel density plots (band width = 3), and BA plots: (A) height, (B) chest, (C) waist, (D) buttock, and (E) hip.** The detailed results are displayed in Table 3 (buttock: the scanner measured the largest circumference around the buttock, the manual measurers measured the circumference on the mid-buttock position).

reflecting the larger circumference and smaller height measurements when scanning the individuals. BMI calculated by mSM was 0.52 kg/m$^2$ (SD = 0.21) higher than calculated by mMM (caused by the lower mSM values for height taken from the scans). The paired $t$-test revealed significant ($p < 0.001$) differences between mSM and mMM for all body shape indices.

**Table 4** Differences between the two methods (mSM vs. mMM) for WHR, WHtR, and BMI when comparing the prevalence for OW/OB according to the official WHO-categories.

| | mSM | | mMM | | Reclassification mSM–mMM (%) | Kappa |
|---|---|---|---|---|---|---|
| | Freq. | % | Freq. | % | | |
| **WC (cm)** | | | | | | 0.60 |
| <94.0 | 114 | 92.7 | 118 | 95.9 | −3.2 | |
| 94.0–101.0 | 9 | 7.3 | 4 | 3.3 | 4.0 | |
| ≥102.0 | 0 | 0.0 | 0 | 0.0 | 0.0 | |
| **WHR** | | | | | | 0.87 |
| <0.9 | 116 | 94.3 | 113 | 91.9 | 2.4 | |
| ≥0.9 | 7 | 5.7 | 9 | 7.3 | −1.6 | |
| ≥1.0 | 0 | 0.0 | 0 | 0.0 | 0.0 | |
| **WHtR** | | | | | | 0.58 |
| <0.5 | 105 | 85.4 | 110 | 89.4 | −4.0 | |
| ≥0.5 | 18 | 14.6 | 12 | 9.8 | 4.8 | |
| **BMI (kg/m²)** | | | | | | 0.76 |
| <18.5 | 2 | 1.6 | 3 | 2.4 | −0.8 | |
| 18.5–24.9 | 92 | 74.8 | 100 | 81.3 | −6.5 | |
| 25.0–29.9 | 28 | 22.8 | 19 | 15.4 | 7.4 | |
| ≥30.0 | 0 | 0.0 | 0 | 0.0 | 0.0 | |

Regardless of the applied method and the calculated body shape index, none of the measured subjects reached the highest risk-related WHO-category of being obese (WC ≥ 102.0 cm, WHR ≥ 1.0, and BMI ≥ 30.0 kg/m²) (Table 4). However, there were differences in the prevalence of overweight between mSM and mMM based calculations within each of the four body shape indices, on the one hand, and between the four body shape indices within one applied method on the other hand. For WC, the prevalence of overweight (94.0–101.0 cm) was 4.0% greater ($N = 5$ out of 123 subjects more) when calculated based on the scans compared to the manual measurements (7.3% vs. 3.3%). Thus, five out of 123 subjects changed from the normal category to the category under risk. For WHR, the differences in the prevalence of overweight (0.9–1.0) between mSM- and mMM-based ratios was the smallest ($-1.6\%$ or $N = 2$) among all body shape indices (5.7% for mSM vs. 7.3% for mMM). When also taking height and not only circumference measurements into account by calculating WHtR and BMI, the prevalence of overweight increased. For WHtR, 14.8% were overweight (WHtR ≥ 0.5) for mSM compared to 9.8% for mMM (difference = 4.8% or $N = 6$). For BMI, 22.8% were overweight (25.0–29.9 kg/m²) for mSM compared to 15.4% for mMM (difference = 7.4% or $N = 9$). Kappa coefficients between scans and manual measurements showed moderate agreement for WHtR (0.58) and WC (0.60), substantial agreement for BMI (0.76), and almost perfect agreement for WHR (0.87).

### Comparing techniques in sub-group analysis regressions

When estimating the associations of sports hours per week, number of visits to a physician during the last year, and age with WC, WHR, WHtR, and BMI we found no significant

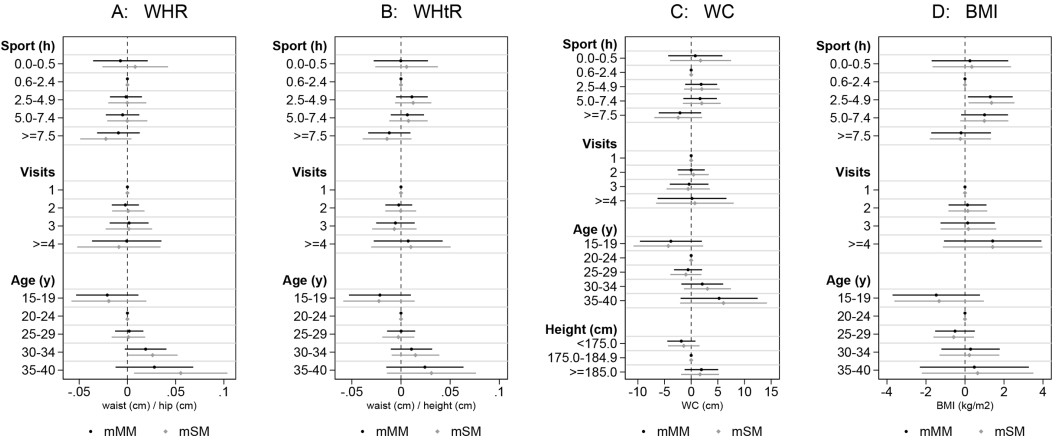

**Figure 3   Comparison between methods: mSM vs. mMM by performing linear regressions (with 95% confidence intervals).** Association of (A) WHR, (B) WHtR, (C) WC, and (D) BMI with hours of sports per week, number of visits to a physician during the last year, and age ($N = 119$, the detailed results are displayed in Table S1).

results for sports hours and visits to a physician (Fig. 3 and Table S1). However, there was a non-significant tendency to lower WC, WHR, and WHtR values among the men performing ≥7.5 h of sports per week. The strongest, but only partially significant association, was found for age, with a tendency for higher values in all body shape indices for men older than 30 years. In general, the performed regressions resulted in very similar coefficients, confidence intervals, and $p$-values among the sub-groups for both, the scan and the manual techniques.

## DISCUSSION

We successfully performed repeated body scans and manual measurements on 123 individuals. We found that the mean differences within the two scan measurements were smaller than within the two manual measurements, and correspondingly precision and intraclass correlation coefficients were higher. We also found both techniques to be highly correlated. Nevertheless, when comparing means between both techniques we found significant differences: height was systematically shorter in the scans, whereas all four circumference measurements taken from the trunk and hip were larger in the scans. In consequence, body shape indices WC, WHtR, and BMI also became significantly larger and, correspondingly, the prevalence of overweight was greater when calculated from the scans. Depending on the index, between 4.1% and 7.3% of the probands ($N = 5$ to 9 men out of 123) changed the risk category from normal to overweight when classified based on the scan measurements. However, when employing regression analyses the two measurement techniques resulted in very similar coefficients, confidence intervals, and $p$-values.

The better precision and repeatability within repeated scan measurements was also found by earlier validation studies on adult probands (*Heuberger, Domina & Macgillivray, 2008*; *Verweij et al., 2013*; *Jaeschke, Steinbrecher & Pischon, 2015*; *Sebo et al., 2015*) and can at least partially be explained by generally reduced inter- and intra-observer variance. On

the one hand, measurement protocols (automatic landmark identification) are more standardised in the scans. On the other hand, the classical manual anthropometric measurements bear more potential for intra- and inter-observer errors in terms of tape-positioning and -tightening, etc. (*World Health Organization (WHO), 2011*). This is also reflected in the more differing appearance of the kernel density plots when comparing two manual circumference measurements against each other in Fig. S2. In particular, manually measuring buttock circumference was subject to larger inter-observer errors in our study. Only height measurements were equally precise in both of the applied techniques. The linear nature of manual height measurement by the means of a fixed stadiometer presumably allows better repeatability (in contrast to tape-measured circumferences).

That height was systematically shorter in the scans than when manually measured conflicts with some of the existing earlier validation studies: In these studies, the larger height values in the scans are explained by issues related to the worn bathing cap (air beneath, or lots of hair up-biasing height) (*Han, Nam & Choi, 2010*; *Jaeschke, Steinbrecher & Pischon, 2015*; *Psikuta et al., 2015*; *Kuehnapfel et al., 2016*). However, our contrary finding could be explained by the following hypothetical factors: In the scanner, participants are standing with the legs hip-wide apart (to enable the scanner/software to correctly identify the crotch), while they are standing with legs closer together when being manually measured (*World Health Organization (WHO), 2011*). Furthermore, the physical contact between the back of the body and the stadiometer in the manual measurement could keep the body posture straight and elongated at the back. In contrast, the probands tend to slightly deviate from the Frankfurt Plane and bend forward in the scans, possibly resulting in a reduced height. We calibrated and controlled our stadiometer several times during the measuring days to eliminate possible confounding technical measurement errors. However, future studies should test these potentially biasing factors.

Our finding that waist, hip, and buttock circumferences were systematically larger in scans than manually measured finds broad support in other validation studies (*Han, Nam & Choi, 2010*; *Choi & Ashdown, 2011*; *Jaeschke, Steinbrecher & Pischon, 2015*; *Ng et al., 2016*). The mean differences we found in the present study were relatively small for waist circumference ($+1.2$ cm), and in the expected range of plus 3–5 cm for hip and chest circumference. When measured manually, the physical contact and proximity to the measurer might influence body posture and contraction of the trunk area of the probands resulting in smaller values (*Jaeschke, Steinbrecher & Pischon, 2015*). In the case of hip circumference, the standard scanning posture with legs hip-width apart might enlarge the resulting measurements (*McKinnon & Istook, 2002*). In general, the hand-held tape measurements have a tendency to compress the circumference caused by too much tension and thus reduce the values (*Wang et al., 2006*). Lastly, the non-automatic positioning of the tape in manual measurements is challenging, even when conducted by trained personnel, which possibly also results in greater variation. In our study, the largest mean difference of 12.6 cm was found for buttock circumference. This unreliably large deviation originates from an unintended inconsistency of measurement positioning between the two techniques: whereas the scanner measured the largest circumference around the buttock, the manual measurers measured

the circumference on the mid-buttock position. This error between protocols highlights the importance to carefully synchronise the manual and scan measurement landmarks.

The smaller height and the larger waist circumference measurements in the scans resulted in higher BMI, WC and WHtR values. For WHR, the systematically greater hip circumference resulted in lower values in the scans. As a consequence, re-classification between the risk categories may become an issue, and the impact on related health outcomes has to be addressed by further studies. However, our data showed relatively low re-classification percentages of 4.1%–7.3% and generally moderate to excellent Kappa agreement results between techniques and risk categories.

Regarding the performed regressions, we found no influence of physical activity or numbers of visits to a physician on all four tested body shape indices, irrespective of the applied technique. However, we found a non-significant tendency to larger body shape with increasing age (again for both techniques), generally reflecting weight-gain with increasing age (*Bogin, 1999*; *Mason & Katzmarzyk, 2009*). More importantly, the two measurement techniques resulted in very similar coefficients, confidence intervals, and *p*-values for the tested sub-groups of age, physical activity, and visits to a physician. This underlines the findings of an earlier study indicating that the two techniques, despite of systematic differences, showed similar correlations to parameters of the metabolic syndrome (*Jaeschke, Steinbrecher & Pischon, 2015*).

This study has several limitations: first, the studied young men are not representative for the general Swiss population. We did not include women and older age groups, who might show deviating differences between the manual and scanned measurements. Randomisation might furthermore be affected due to the voluntary participation of the probands, which presumably resulted in a lower than average prevalence of overweight and obese individuals in the sample. In particular, the differences between the two techniques might be larger in obese individuals, even when a recent validation study (*Kuehnapfel et al., 2016*) did not find significant diverging applicability results for obese probands. On the one hand, the homogeneous composition of our relatively large sample allowed for a comparison of the two techniques without possibly biasing body shape extremes (very short/tall or obese probands can affect the variability of tape measurements (*Gibson, 2005*)). On the other hand, epidemiological studies often cover a wide variability of body shapes. To enhance the representativeness such validation studies should rely on samples with greater heterogeneity and also include obese individuals. Second, the applied questionnaire was limited in its explanatory power. In the non-clinical setting of our study we were not allowed to collect more detailed health information or even measure parameters of the metabolic syndrome. However, testing determinants of body shape was not the main interest of this study, as we aimed at comparing the two techniques upon differences in the effect size of selected determinants. Third, anthropometric measures are also influenced by the time of the day (e.g., being taller in the morning than in the evening) or the training level of the measurers. We made an effort to reduce these possible sources of error by taking scans and manual measurements isochronally within 20 min for each participant and by additionally briefing the experienced medical measurers to strictly stick to the protocols (*Sebo et al., 2015*). Fourth, we did not have access to the software algorithms calculating the 3D virtual
body and consecutively the measurements from it. As for earlier validation studies, these algorithms were not accessible (*Kuehnapfel et al., 2016*) and could not be controlled for their possible influence on the results of this study. Fifth, we only compared five selected of the over 150 available standard scan measurements with their manual equivalent. However, other studies have shown the validity of the scans also for other standard measurements, even when particular body areas such as the top of the shoulders or the neck area are more difficult to assess by the scans (*Daniell, Olds & Tomkinson, 2012*; *Tomkinson & Shaw, 2013*).

Our study highlights the strengths and limitations regarding the use of 3D technology in large-scale surveys intended to categorise nutritional status for public health ends, as opposed to obtaining data for the retail industry which has been the main purpose of large 3D surveys to date. The high correlations between 3D and manual measurements indicate that 3D technology is very successful at ranking individuals within the population. Nevertheless, the biases between techniques demonstrate problems in categorising with precision whether a specific individual has crossed the threshold for underweight, overweight or obesity. If future standardisation could reduce the between-subject variability in this bias, for example through greater standardisation of posture, etc., it could be possible to introduce correction factors removing the average bias. Further work will be necessary to test this in greater detail.

## CONCLUSION

We can confirm other validation studies in that the scans generally showed good feasibility, reliability, and validity when used to perform a large number of measurements from a larger number of probands in a short time. On the one hand, scanned and manually measured anthropometric data are highly correlated, but—on the other hand—there also exist important systematic differences due to the nature of the techniques (contact vs. non-contact; different body positions). Thus, the two techniques are not directly equivalent, which makes the comparison between averages (e.g., to show time trends) from different techniques impossible without developing correction factors. Nevertheless, the non-invasive and fast character, the low potential for measurement errors, the high reproducibility, and the potential to reassess the archived virtual bodies—even years later by improved software or new algorithms—make the technique attractive for application in an epidemiological setting.

In order to improve the acceptance of the scan technique, further studies should develop a universally accepted standard scan measurement protocol (focussing on posture, breathing position, etc.), precisely assess the systematic bias between the techniques and its causes in larger samples or metastudies (in particular also for height), address the impact of reclassification among OW/OB risk groups, apply the scanner in follow-up studies to trace body shape changes of an individual over time, and also to optimise the software algorithms in order to correct for the systematic differences between techniques as well as to develop new body shape models or indicators that are even more closely related to health outcomes than the standard anthropometric approaches in use.

## ACKNOWLEDGEMENTS

The authors wish to thank: Armasuisse for providing their 3D Scanliner; Andreas Stier, and Thomas Mathis (from Armasuisse) for their technical and scan measurement support; Lena Öhrström, Karl Link, Roger Seiler, and Joël Floris (all IEM, University of Zurich) for their manual measurement assistance; Karl Rüesch, Ralph Stöber, and René Zimmermann (from the University of Zurich) for their technical support; Helga Gäbel (Human Solutions GmbH, Kaiserslautern, Germany), Tobias Pischon (MDC, Berlin), Dörte Radke (Greifswald University), Peter Ahnert (Leipzig University), Peter Jüni (University of Toronto), and Barry Bogin (Loughborough University) as well as Daniel Tiller (Halle University and University Hospital) and two other reviewers for their helpful comments.

### Funding

This study was funded by Mäxi Foundation, Zurich, Switzerland (F. Rühli). The funders had no role in study design, data collection and analysis, decision to publish, or preparation of the manuscript.

### Grant Disclosures

The following grant information was disclosed by the authors:
Mäxi Foundation.

### Competing Interests

The authors declare there are no competing interests.

### Author Contributions

- Nikola Koepke analyzed the data, wrote the paper, prepared figures and/or tables, reviewed drafts of the paper.
- Marcel Zwahlen conceived and designed the experiments, analyzed the data, prepared figures and/or tables, reviewed drafts of the paper.
- Jonathan C. Wells conceived and designed the experiments, performed the experiments, analyzed the data, reviewed drafts of the paper.
- Nicole Bender analyzed the data, reviewed drafts of the paper.
- Maciej Henneberg conceived and designed the experiments, performed the experiments, reviewed drafts of the paper.
- Frank J. Rühli conceived and designed the experiments, performed the experiments, contributed reagents/materials/analysis tools, reviewed drafts of the paper, funding.
- Kaspar Staub conceived and designed the experiments, performed the experiments, analyzed the data, contributed reagents/materials/analysis tools, wrote the paper, prepared figures and/or tables, reviewed drafts of the paper, study Superviser.

### Human Ethics

The following information was supplied relating to ethical approvals (i.e., approving body and any reference numbers):

All of the young men participated voluntarily in this study. They were informed in detail by the study leaders about the study (aim, protocol, risks, etc.) in written form prior to the measurement days and once again orally at the beginning of their measurements by the study leaders. Prior to the start of the measurements all participants signed a detailed informed consent form. At the time of measurement (February 2013) a waiver by the ethics committee of the Canton of Zurich was not needed for such non-clinical and non-invasive studies.

## Data Availability

The raw data has been supplied as a Supplementary File.

## Supplemental Information

Supplemental information for this article can be found online at http://dx.doi.org/10.7717/peerj.2980#supplemental-information.

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
