# Peer review of "Comparison of 3D laser-based photonic scans and manual anthropometric measurements of body size and shape in a validation study of 123 young Swiss men"

_PeerJ, doi:10.7717/peerj.2980_

## Round 0.1 · original submission · Major Revisions

Please ensure you address all of the reviewers' comments in the revised article. If you do not agree with a criticism, please provide a rebuttal.

Please also address these issues:

The correlation coefficient is not a good measure of agreement as it can still be high despite a considerable systematic difference (as demonstrated in this study). The better way would be to use Lin’s concordance correlation coefficient (Lin 1989), which adjusts Pearson’s r by a bias correction factor. There is a Stata command by Nick Cox to calculate the CCC (use “ssc install concord” within Stata).
Lin LI. A concordance correlation coefficient to evaluate reproducibility. Biometrics. 1989; 45: 255-68.

Have the investigators tried to perform the height measurement in the scanner with legs close together (in addition to legs apart) and used the former to measure height? Is there some data to back that claim in lines 384-6?

Harmonize the spelling in the manuscript: It should be either British or US but not both. It appears to be leaning towards British spelling so it’s probably easiest to replace “standardized” with “standardised” throughout (I may have missed other US spelled words)

Lines 223-224: Please correct the BMI formula (kg/m2)
Line 266: replace “waver” with “waiver”
Line 344: replace “insignificant” with “non-significant”
Line 345: replace “clearest” with “strongest”
Line 379: replace “contradicts” with “conflicts with”
Line 439: please reword as there can’t be confounding with this study design
Line 446: replace “training state” with “experience” or “training level”
Line 446: replace “We took effort” with “We made an effort”
Line 447: again, confounding requires an exposure-outcome relationship; these are rather sources of random error
Figure 3: Could the authors please somehow increase the spacing between the black and gray bars? In its current form the plots look very blurry/3D-ish due to the close proximity of the two bars.

·

Basic reporting

Obesity and overweight are important risk factors for morbidity and mortality. Therefore, the reliable measurement of the anthropometric measures is crucial to determine the risk of chronic diseases such as the metabolic syndrome.
However, the accurate measurement of anthropometric markers in population based epidemiological studies is challenging due to a high inter- and intra-observer variability of the manually derived measures. 3D body scanning technology promises to measure accurately a large number of body measures within a few seconds.
The present paper from Koepke et.al. aims to compare the scanning and the manual anthropometric measurement techniques based on five selected body measurements.

Overall, the paper is very well written and easy to follow. The study adds valuable data to the still small number of validation studies comparing the scan technique with manual measurements. All relevant studies have been taken into account in the introduction section as well as in the discussion section. The conclusions drawn from the results are mostly traceable and coherent. Tables and figures are presented in an appropriate manner.
However there are some points need to be addressed:

The words “cross-sectional study” should be replaced by “validation study” in the title.

Experimental design

The main limitation is the selection of the study participants, restricted to young men aged 18 to 38 years. Furthermore, there were no obese subjects included in the study. Our (still unpublished) analysis showed greater differences between manually derived anthropometric measures in women compared with men. Furthermore, our data indicates greater differences in obese subjects (with BMI greater or equal 30 kg/m2) compared with subjects with a BMI lower than 30 kg/m2. On the one hand, a low variability might facilitate the comparison of the two techniques. On the other hand, in epidemiological studies we see a wide variability of body shapes. Therefore, to make the results more representative validation studies should include samples with greater heterogeneity. This should be further addressed in the discussion section.

Can the authors explain the selection of the selected body scanner measures? For instance, “high waist girth” seems to be a more precise automatic measure for waist circumference than “waist girth”. Furthermore, chest circumference is an uncommon marker for general or abdominal obesity in epidemiological studies. The authors should explain more detailed the selection of the studied measures.

Validity of the findings

Could you exclude a bias for the comparison of the two manually derived measurements? I would expect that the examiner remembers the afore measured value so the subsequent measurement could be biased.

The findings regarding the association of health determinants and anthropometric measures are not very meaningful due to the limited information on the health status of the study participants (threats of confounding) and the selected sample (threats of bias). Therefore, I recommend omitting this analysis in the present publication.

Reviewer 2 ·

Basic reporting

1.Table 1: Physician number of visit non of interest for the study, add height and BMI
2. Specify abbreviations (perc= %?)
3. Table 2 on manual measuements) : take out height
4. Table 4: take out obesity parameters Waist > 102, BMI > 30 (not of interest with your data)

Experimental design

*tape measurements (body circumferences) and height should be done 3 times.
*any comment about weight measurement
*buttock circumference is of no interest for clinical outcomes. I'd like to know why they chose it.

Validity of the findings

1. Concern: is that type of 3D scan validated for body composition? (i couldn't find anything about it)
2. tape measurement are not influenced by the time of the day (line 445)

Additional comments

I think it's a good idea as a study however, in order to have clinical meaning, I'd add comparison of normal to OW measurement regarding all parameters like in Olivares J, Wang J, Yu W, Pereg V, Weil R, Kovacs B, Gallagher D, Pi-Sunyer FX. Comparisons of body volumes and dimensions using three-dimensional photonic scanning in adult Hispanic-Americans and Caucasian-Americans. J Diabetes Sci Technol. 2007 Nov;1(6):921-8.

Reviewer 3 ·

Basic reporting

Overall format is adherent to policies. Below are minor comments.
1. Table 2: Please add the units of precision?
2. Line 70: replace comma with semicolon (two independent clauses)
3. Line 128: conjunction “and” after the semicolon is unnecessary here

Experimental design

1. This is a fast moving field and there have been some recent articles that should be referenced, inlcuding BK Ng et al. “Clinical anthropometrics and body composition from 3D whole-body surface scans” and Lee JJ et al. “Prediction of android and gynoid body adiposity via a three-dimensional stereovision body imaging system and dual-energy X-ray absorptiometry.” The results here should be contrasted with these papers in terms of accuracy and precision.
2. V.B. The comments about the use of the body volume estimates using Siri need to include caveates that body composition from total body volume is highly unstable and that small differences in body volume can create large errors on body composition.
3. A parametric test, such as an ANOVA, should be estimated to evaluate the precision of each of the measurement techniques. Lee JJ et al. shows techniques for comparisons.
4. Table 3: The WHR correlation seems very so low. Please justify.
5. Figure 2: Include how the buttocks are defined differently for SM and MM?
6. Figure 3: Please explain how waist circumference varies by hours of sport does not add much value to the main purpose of this article, which is to compare two measurement techniques.
7. Appendix Figure 2: Please explain why the kernel density plots appear so differently in the buttocks region?

Validity of the findings

1. The population is incompletely described. No mention is given to male/female ratio, where this sample came from, or how they were recruited? Were they a healthy cohort? Unknown health status? Please include more detail.
2. Please provide what quality assurance was performed on the scanner to insure accuracy over the course of the study.

Additional comments

This manuscript evaluated the accuracy and precision of a 3D laser-based scan (SM) against manually-derived (MM) anthropometric measures. The motivation is to add to the collective knowledge on this relatively new field of optic-based scanning of anthropometric measures as indicators of health. The study included 123 young male participants, and their data were collected in February 2013. Each participant underwent 3 tests: a questionnaire, a 3D laser scan, and a manual measurement of anthropometric variables defined by the World Health Organization. The study showed that most measures were precise for both SM and MM (except for the chest, waist, and buttock for MM). The study also showed that the two techniques were highly correlated and, thus as expected, similar values for linear regression.

---

## Round 0.2 · accepted · Accept

The authors have satisfactorily addressed the reviewers' comments.